# Investigation of the Persistence, Toxicological Effects, and Ecological Issues of S-Triazine Herbicides and Their Biodegradation Using Emerging Technologies: A Review

**DOI:** 10.3390/microorganisms11102558

**Published:** 2023-10-13

**Authors:** Sajjad Ahmad, Murugesan Chandrasekaran, Hafiz Waqas Ahmad

**Affiliations:** 1Environmental Sustainability & Health Institute (ESHI), City Campus, School of Food Science & Environmental Health, Technological University Dublin, Grangegorman Lower, D07 EWV4 Dublin, Ireland; 2Key Laboratory of Integrated Pest Management of Crop in South China, Key Laboratory of Natural Pesticide and Chemical Biology, Ministry of Agriculture and Rural Affairs, Ministry of Education, South China Agricultural University, Guangzhou 510642, China; 3Department of Entomology, Faculty of Agriculture, University of Agriculture, Faisalabad 38000, Pakistan; 4Department of Food Science and Biotechnology, Sejong University, Neungdong-ro 209, Seoul 05006, Republic of Korea; chandrubdubio@sejong.ac.kr; 5Department of Food Engineering, Faculty of Agricultural Engineering & Technology, University of Agriculture, Faisalabad 38000, Pakistan; hafiz_waqasahmad@yahoo.com

**Keywords:** s-triazine herbicides, toxicity, enzymes and genes, molecular mechanism, phytoremediation

## Abstract

S-triazines are a group of herbicides that are extensively applied to control broadleaf weeds and grasses in agricultural production. They are mainly taken up through plant roots and are transformed by xylem tissues throughout the plant system. They are highly persistent and have a long half-life in the environment. Due to imprudent use, their toxic residues have enormously increased in the last few years and are frequently detected in food commodities, which causes chronic diseases in humans and mammals. However, for the safety of the environment and the diversity of living organisms, the removal of s-triazine herbicides has received widespread attention. In this review, the degradation of s-triazine herbicides and their intermediates by indigenous microbial species, genes, enzymes, plants, and nanoparticles are systematically investigated. The hydrolytic degradation of substituents on the s-triazine ring is catalyzed by enzymes from the amidohydrolase superfamily and yields cyanuric acid as an intermediate. Cyanuric acid is further metabolized into ammonia and carbon dioxide. Microbial-free cells efficiently degrade s-triazine herbicides in laboratory as well as field trials. Additionally, the combinatorial approach of nanomaterials with indigenous microbes has vast potential and considered sustainable for removing toxic residues in the agroecosystem. Due to their smaller size and unique properties, they are equally distributed in sediments, soil, water bodies, and even small crevices. Finally, this paper highlights the implementation of bioinformatics and molecular tools, which provide a myriad of new methods to monitor the biodegradation of s-triazine herbicides and help to identify the diverse number of microbial communities that actively participate in the biodegradation process.

## 1. Introduction

Since the 1950s, synthetic s-triazine herbicides (e.g., atrazine, atraton, ametryn, cyanazine, prometryn, prometon, propazine, simazine, simetryn, terbutryn, trietazine, terbuthylazine, metribuzin, terbumeton, simeton, and desmetryn) have been extensively applied in the agriculture sector for the control of broadleaf weeds and annual grasses all over the globe [1] (Figure 1). Besides this, these compounds and their derivatives are excessively applied in other industrial applications such as the chlorine disinfection process, electrical varnishes, and the reduction of nitrogen oxides in stationary diesel engine exhaust gases, and as dyes, pharmaceuticals, explosives, polymer intermediates, and fire retardants [2]. Some have been banned for application in developed countries due to their high risk of pollution in the agroecosystem [3]. These herbicides are taken up via plant roots and by xylem tissues circulated throughout the plant system. Triazines act by inhibiting photosynthesis in leaves, particularly impeding the photosystem. They can absorb various fruits’ and vegetables’ surface skin and enter their flesh part [4].

The imprudent application of this group of herbicides in agriculture and forestry means that their residues are frequently detected in water and food crops such as fruits and vegetables. Moreover, their residues are also detected in various food-processed products such as fruit juices [5]. The typical s-triazine herbicides, such as atrazine, simazine, and propazine, generated sub-lethal effects on another non-organisms (Figure 2). The hazardous residues of these compounds cause many chronic diseases in living organisms, such as the fluctuation of hormones, genotoxicity, cytotoxicity, reduced birth rate, and reproductive diseases in different vertebrates, including mammals and humans [6]. More alarmingly, the imprudent use of triazines affected beneficial flora and fauna, such as changing the flowering and blooming stages and collapsing the colony order of pollinators and predators. Additionally, a large number of triazine herbicides prevent electron transport in the photosynthesis process [7]. Recently, Tian et al. [8] established a simple, reliable, cost-effective, and eco-friendly method for the simultaneous detection of s-triazine herbicides, especially atrazine, prometryn, and simetryn from honey samples. Their results showed that the maximum residual limit of all herbicides was 0.1 to 1.5 µg/kg.

It has been shown that more than 70% of residues of s-triazines enter the aquatic system through leaching or underground surface runoff [9]. These hazardous substances threaten aquatic creatures such as fish, amphibians, crabs, beneficial algae, and other micro and macro marine organisms [10,11]. Recent research explained that the residues of triazine herbicides are found at high ratios in different muscles and other body parts, such as the liver, of birds, mammals, and fish [12]. To elevate the agroecosystem toxicity of s-triazine herbicides, efficient, less costly, attractive, eco-friendly, and sustainable techniques are immediately required to remove these hazardous compounds. In general, the removal of various toxic pollutants and their transformation into less toxic substances from the ecosystem can be achieved via different methods and their associated techniques, such as physical adsorption, the advanced oxidation process, chemical remediation, photoremediation, phytoremediation, and biodegradation [13]. Among all the techniques, biodegradation using indigenous microbial species to remediate s-triazine herbicides and their hybridization with nanomaterials and plants (phytoremediation) is considered more effective, low-cost, eco-friendly, and more prominent [14].

For decades, many microbial species have been isolated from the environment, making it feasible to degrade different types of hazardous compounds to preserve the environment [15]. Until early 1990, numerous studies revealed that the bacterial metabolism of s-triazine herbicides, especially for atrazine in an open field environment, occurs through N-dealkylation of the s-triazine ring and does not typically lead to ring cleavage [16,17]. Later on, since 1995, many microbial strains that degrade s-triazine herbicides via the hydroxylation process, followed by the direct removal of alkylamino groups, have been investigated [18]. More interestingly, these microbial strains were reported to possess more effective properties, such as the complete mineralization of s-triazines and their toxic metabolites, validating the approach of microbial strains to degrade xenobiotics at a higher degradation rate for future practical applications [19]. Besides the use of microbial strains to achieve the efficient bioremediation of s-triazines, various microbial enzymes and genes provide fundamental insights to recognize transformation pathways to less toxic products [20].

To the best of our knowledge, the current study is the first to report on s-triazine herbicides. It mainly focuses on the toxicity, persistence, and biodegradation of s-triazine herbicides. This study highlights potential enzymes and genes that impart foundational insights for understanding the transformation pathway and ultimately influencing the efficacy of the biodegradation strategy. Additionally, we focus on the recent development of microbial active-element-centered techniques with material science and plant science using modern biological molecular tools, which establish a myriad of promising strategies to promote the potential for environmental monitoring and improving the in situ degradation of s-triazines. Also, we focus on the recent developments in nanotechnology and the synthesis of nanoparticles via physical, chemical, and biological methods and their conjugation with whole cells, enzymes, and genes, which create a myriad of encouraging techniques to enhance the degradation rate of s-triazine herbicides and other agrochemicals and xenobiotics such as dyes, heavy metals, microplastics, and other emerging micropollutants for environmental monitoring and their conservation. This review offers additional information to that provided by other published studies of nanotechnology, molecular biology, and microbial agents in environmental cleanup. Our work will interest many readers, including environmentalists, biologists, agronomists, and materials scientists.

## 2. Potential Microbial Species Employed for Biodegradation of S-Triazines

Microbes play a vital role in the clean-up of various kinds of contaminants from the environment because they can survive and withstand stress in several ways [21]. The employment of microbial species to degrade various pollutants (heavy metals, dyes, pesticides, hydrocarbons, antibiotics, and other emerging pollutants) has received a wide range of attention [22]. More interestingly, the degradation of pesticides using microbial species usually did not generate any secondary toxic substances after mineralization and was shown to be its crucial feature [23]. Besides this, using a co-metabolism mechanism, indigenous microbial species effectively degraded toxic compounds into less toxic substances, which they can further utilize as a carbon and energy source [15].

The degradation rate and activation of microbial species in a contaminated place are dependent on various biotic and abiotic factors such, as the concentration of hazardous compounds, their structure and solubility, the kind of soil, moisture content, temperature, pH, soil nutrients, and organic matter [24,25]. A small unit of one cubic meter collected from any topsoil potentially has millions of microorganisms, including fungi, approximately ten times more bacteria and actinomycetes, slightly fewer nematodes, thousands of macroscopic earthworms, and innumerable protozoa and algae [26]. Furthermore, they can secrete various intracellular and extracellular enzymes, targeting contaminants accumulated in the environment and altered into their derivatives, which can further be used by microbial genera [27]. However, microbial communities play a crucial role in the bio-purification system and clean up the polluted environment efficiently (Figure 3).

The unwise use of s-triazine in various sectors has received serious attention from researchers worldwide due to its high toxicity and acute persistence in the ecosystem and human health. In the s-triazine group, atrazine is widely applied to control grass and broadleaf weeds in cash crops such as sugarcane, wheat, maize, and rice [28,29]. Due to its long half-life (42–231 days) and low adsorption ability in agricultural soil, it not only pollutes the soil, but its residues also penetrate groundwater and are frequently detected in drinking water [30]. In previous studies, numerous indigenous and non-indigenous microbial species were reported that rapidly degraded s-triazine herbicides from the polluted environment (Table 1).

Recently, Cao et al. [31] isolated a bacterial strain, *Arthrobacter* sp. C2, from atrazine-contaminated soil. After genomic identification, different parameters and degradation characteristics were analyzed for atrazine degradation. The genomic results indicated that this bacterial strain comprises 4,305,216 base pair nucleotides, three plasmids, and 4705 coding genes. Meanwhile, the degradation results revealed that this bacterial strain could survive more efficiently at 30 °C and at a pH range of 7–9, and degraded atrazine from low to high concentrations at 1, 10, and 100 mg/L with degradation rates of 0.34, 1.94, and 18.64 mg/L per day, respectively. Last, in the same study, the metabolic pathway of atrazine was proposed, and the authors identified four metabolites named hydroxyatrazine, N-isopropylammelide, deisopropylhydroxyatrazine, and cyanuric acid. This study concluded that the bacterial strain C2 is a potential agent for the biodegradation of atrazine in the agroecosystem. In another study, Chen et al. [32] isolated a native microbial strain, *Paenarthrobacter* sp. W11, from the soil to degrade atrazine smoothly by adopting bioaugmentation and biodegradation techniques in wheat crops, and the alteration of microbial species during the degradation mechanism was analyzed. Their results demonstrated that the bioaugmentation process efficiently enhanced the degradation rate of atrazine in soil and reduced the toxic effects of residues on wheat growth. The findings of this study also revealed that *Methylobacillus* sp. and *Pseudomonas* sp. played a more crucial role in the biodegradation process because their relative abundance significantly matched the relative abundance of the strain *Paenarthrobacter* sp. W11 and the atrazine mineralization pathway. This study provides new insight into the collaboration of native microbial species and external inoculums for removing atrazine. Zhu et al. [33] isolated a bacterial strain from the soil and, based on 16 rRNA and morphological and physiological analysis, identified it as *Bacillus atrophaeus* YQJ-6 for the effective removal of atrazine from the ecosystem by investigating a variety of parameters, such as the effect of temperature, initial concentration, and effect of pH. The results of this study explained that strain YQJ-6 efficiently degraded atrazine 99.2% at a concentration of 50 mg/L and a wide range of temperatures (30–40 °C) and pH (7–7.5). A tolerance test of the bacterial strain was also conducted, and the findings showed that this strain could tolerate 1000 mg/L atrazine.

A bacterial strain, *Arthrobacter ureafaciens* XMJ-Z01, was isolated from soil and its degradation ability tested against simazine. The results showed that this strain revealed high resistance and tolerated simazine up to a concentration of 2000 mg/L, and within a week, was able to degrade 99.1% at a concentration of 100 mg/L. Besides this, its effect on accelerating the degradation ability of soil enzymes was tested by adding less fresh soil to the culture medium. The results demonstrated that this strain reduced the toxicity and adverse effects of simazine on soil enzymes [34]. In another study, a bacterial strain, *Arthrobacter urefaciens*, was screened, and their degradation ability for simazine was examined. Using this bacterial strain, 100% degradation of simazine was achieved in four days [35]. The complete degradation of ametryn at a concentration of 2 mg/L was achieved in 5 days by the bacterial strain *Nocardioides* DN36 [18]. Guo et al. [36] isolated a bacterial strain from contaminated soil and identified it as *Arthrobacter* sp. SD3-25, which efficiently degraded simazine toxic residues from the environment. They found that external carbon sources such as sucrose enhance the quantity of simazine-degrading genes (*trzN*, *atzB,* and *atzC*) in bacteria-amended soil. Moreover, this bacterial stain effectively degraded simazine and enhanced microbial communities in soil.

**Table 1 microorganisms-11-02558-t001:** Functional microbial species involved in the effective biodegradation of s-triazine herbicides.

Target S-Triazine Herbicide	Microorganisms	Source of Isolation	Degradation (%)	References
Atrazine	*Acinetobacter* sp. Strain A6	Polluted soil	80	[37]
	*Rhodococcus* sp. BCH2	Polluted soil	75	[38]
	*Bacillus badius* ABP6	Maize field	89.7	[39]
	*Arthrobacter* sp. C3	Corn field	100	[40]
	*Cryptococcus laurentii*	Agricultural soil	100	[41]
	*Klebsiella variicola* FH-1 *Arthrobacter* sp. NJ-1	Polluted soil	97.4	[42]
	*Ochrobactrum oryzae*	Wastewater	83.5	[43]
	*Rhodobacter sphaeroides* W16	Agricultural soil	96.86	[44]
	*Penicillium* sp.	Corn field	50	[45]
	*Pseudomonas* sp. ADP	Commercial	99	[46]
	*Citricoccus* sp. strain TT3	Wastewater	100	[47]
	*Alcaligenes* sp. S3	Agricultural field	90.56	[48]
	*Arthrobacter* sp. DNS10 *Enterobacter* sp. P1	Polluted soil	99.18	[49]
	*Pseudomonas* sp.	Agricultural soil	99	[50]
	*Pleurotus ostreatus* INCQS 40310	Commercial	82	[51]
Atraton	*Leucobacter* sp. JW-1	Polluted soil	98	[52]
	*Leucobacter* sp. JW-1	Polluted soil	13	[53]
	*Pseudomonas* sp. V1	Wastewater	10	[54]
Ametryn	*Chryseobacterium* sp. *Variovorax* sp., *Aeromonas* sp. and *Xanthobacter* sp.	Agricultural soil	97	[55]
	*Rhodococcus* sp. FJ1117YT	Agricultural soil	50	[56]
	*Scenedesmus vacuolatus*	Wastewater	94.7	[57]
	*Acidithiobacillu ferrooxidans*	Acid mine drainage	84.9	[58]
	*Arthrobacter* sp. MCM B-436	Rhizosphere soil	92	[59]
	*Acidithiobcillus ferrooxidans* BMSNITK17	Soil	94.24	[60]
	*Pseudomonas* sp. YAYA6	Soil	50	[61]
	*Pseudomonas* sp. DY-1	Paddy field	87.8	[62]
	*Rhodococcus* sp. FJ1117YT	Soil	48.9	[56]
	*Nocardioides* sp. DN36	Paddy field	100	[18]
	*Leucobacter* sp. JW-1	Soil	99.9	[52]
Cyanazine	*Agrobacterium radiobacte* M91-3	Commercial	100	[63]
	*Arthrobacter nicotinovorans* HIM	Agricultural soil	100	[64]
	*Rhodococcus corallinus, Pseudomonas* sp. D	Soil	99	[65]
	*Rhodococcus* TE1	Soil	100	[66]
	*Acinetobacter* sp. A6	Agricultural soil	100	[67]
	*Arthrobacter* sp. MCM B-436	Rhizosphere soil	97	[59]
Desmetryn	*Pseudomonas* sp. DY-1	Paddy field	93.6	[62]
	*Rhodococcus* sp. FJ1117YT	Soil	63.9	[56]
Dimethametryn	*Rhodococcus* sp. FJ1117YT	Agricultural soil	100	[68]
	*Bacillus cereus* JUN7	Soil	100	[69]
	*Rhodococcus* sp. FJ1117YT	Soil	81.1	[56]
	*Nocardioides* sp. DN36	Paddy field	100	[18]
	*Pleurotus mutilus*	Commercial	70	[70]
	*Burkholderia cepacia* CH-9	Soil	86	[71]
	*Botrytis cinerea*, *Sordaria superba*, *Absidia fusca*	Wastewater	50	[72]
	*Paracoccus* sp. QCT6	Polluted soil	86.4	[73]
	*Bacillus subtilis*	Agricultural soil	98	[74]
	*Pseudomonas* sp. DY-1	Paddy field	38.2	[62]
	*Rhodococcus* sp. FJ1117YT	Soil	75.5	[56]
	*Nocardioides* sp. DN36	Paddy field	100	[18]
Prometryn	*Microbacterium* sp., *Enterobacter* sp., *Acinetobacter* sp., and *Flavobacterium* sp.	Soil	100	[75]
	*Rhodococcus* sp. FJ1117YT	Soil	75.5	[56]
	*Nocardioides* sp. DN36	Paddy field	100	[18]
	*Leucobacter* sp. JW-1	Soil	100	[52]
	*Aspergillus* sp.	Commercial	61	[76]
	*Leucobacter* sp. JW-1	Soil	100	[53]
	*Pseudomonas* sp. DY-1	Agricultural soil	100	[62]
Prometon	*Leucobacter* sp. JW-1	Soil	95.2	[52]
	*Leucobacter* sp. JW-1	Soil	10	[53]
Propazine	*Leucobacter* sp. JW-1	Soil	100	[52]
	*Pseudomonas stutzeri* sp. Y2	Commercial	96	[77]
	*Arthrobacter* sp. MCM B-436	Rhizosphere soil	87	[59]
	*Leucobacter* sp. JW-1	Soil	41	[53]
	*Pleurotus ostreatus* INCQS 40310	Commercial	90	[78]
	*Phanerochaete chrysosporium*	Commercial	100	[79]
Simazine	*Klebsiella pneumoniae*	Soil	100	[80]
	*Arthrobacter* sp. MCM B-436	Rhizosphere soil	73	[59]
	*β-proteobacterium* CDB21	Bacterial consortium	100	[81]
	*Bradyrhizobium japonicam* CSB1, *Arthrobacter* sp. CD7w and *β-Proteobacteria* CDB21	Agricultural soil	100	[68]
	*Leucobacter* sp. JW-1	Soil	77.9	[52]
	*Phanerochaete chrysosporium*	Commercial	100	[79]
	*Arthrobacter ureafaciens* XMJ-Z01	Soil	99.1	[34]
	*Leucobacter* sp. JW-1	Soil	28	[53]
	*Pseudomonas stutzeri* sp. Y2	Commercial	100	[77]
Simetryn	*Pseudomonas* sp. DY-1	Paddy field	90.2	[62]
	*Rhodococcus* sp. FJ1117YT	Agricultural soil	100	[68]
	*Nocardioides* sp. DN36	Paddy field	100	[18]
	*Rhodococcus* sp. FJ1117YT	Soil	100	[56]
	*Leucobacter* sp. JW-1	Soil	100	[53]
	*Bacillus cereus* JUN7	Soil	100	[69]
	*Leucobacter* sp. JW-1	Soil	77.9	[52]
Simeton	*Nocardioides* sp. DN36	Paddy field	100	[18]
Terbuthylazine	*Leucobacter* sp. JW-1	Soil	98.9	[52]
	*Pseudomonas stutzeri* sp. Y2	Commercial	100	[77]
	*Phanerochaete chrysosporium*	Commercial	100	[79]
Terbumeton	*Leucobacter* sp. JW-1	Soil	31.6	[52]
	*Leucobacter* sp. JW-1	Soil	12	[53]

Microbial degradation summary: This table contains a list of microbial species that are likely to be involved in the biodegradation of s-triazine herbicides. Microbial biodegradation is an important process in environmental remediation, as certain microorganisms have the ability to break down herbicides and other pollutants into less harmful compounds. This table includes information about the species names, source of isolation, target compound, and their effectiveness in breaking down s-triazine herbicides.

Gopal et al. [71] used a potential bacterial strain, *Burkholderia cepacia* CH9, which was screened for its degradation of the metribuzin herbicide. They found that the bacterial strain efficiently degraded 86% metribuzin at a concentration of 50 mg/L in a mineral salt medium within 20 days. Recently, Wahla et al. (2020) [82] used immobilized metribuzin-degrading bacteria (*Rhodococcus rhodochrous* AQ1, *Bacillus tequilensis* AQ2, *Bacillus aryabhattai* AQ3, and *Bacillus safensis* AQ4) on biochar and evaluated their degradation efficiency in the lab as well as in a field-scale environment in a tomato field. Their results showed that potato-vegetated soil-free bacterial cells could degrade 82% of metribuzin, while an immobilized consortium degraded 96%. Moreover, this study clarified that the immobilized microbial consortium was involved in plant growth and restoring indigenous soil microbial communities and soil nutrients. Further, an indigenous potential bacterial strain, *Nocardioides* sp. ATD6, isolated from paddy soil showed efficient degradation of melamine, converted into ammeline and ammelide, and accumulated as cyanuric acid and ammonium [83]. Similar results were found for the soil bacterium *Microbacterium esteramaticum* MEL1, which is considered a superior agent for the degradation of melamine and its metabolites, such as ammeline, ammelide, and cyanuric acid [84]. Zhang et al. [77] isolated *Pseudomonas stutzeri* Y2 from contaminated soil and used it for the degradation of a group of s-triazine herbicides with the help of different immobilized carriers. Their results revealed that the immobilized bacterial strain completely degraded atrazine, simazine, terbuthylazine, and 96% of propazine in 4 days at the initial 50 mg/L concentration. But they did not find that it had degradation efficiency against ametryn, prometryn, atraton, and prometon. This research concluded that strain Y2 was a prominent agent that can degrade not only chlorine-containing s-triazine herbicides but also the other types of s-triazine herbicides that contain methylthio groups, like ametryn and prometryn, and methoxy groups, such as atraton and prometon. In another study, cells of the bacterial strain *Leucobacter* sp. JW-1 were immobilized in polyvinyl alcohol and sodium alginate carrier materials to achieve better degradation results. The degradation experiment results demonstrated that immobilized beads effectively degraded 99.9%, 99.9%, 97.8%, 100%, 77.9%, 98.9%, 95.2%, 98.9%, and 31.6% of atrazine, ametryn, propazine, simetryn, simazine, terbuthylazine, prometon, atraton, and terbumeton, respectively, at an initial concentration of 50 mg/L. This study concluded that immobilized bacterial cells could degrade herbicides more rapidly than free cells and could be reused again [52]. Radosevich et al. [85] isolated soil bacterium M91-3 and found that this strain grew more rapidly in glucose enrichment media and was able to degrade atrazine, cyanazine, and simazine.

In addition to bacterial species, fungal strains are also extensively applied for the remediation of s-triazine herbicides and other xenobiotics from the agroecosystem due to promising characteristics such as the secretion of intracellular and extracellular enzymes [86]. Fungal strains contain mycelia that facilitate intense diffusion into the agroecosystem and provide better surface area for the absorption of xenobiotics [87]. All these properties of fungal species are considered prominent tools for cleaning up environmental pollutants [88].

Recently, Esparza-Naranjo et al. [89] studied the degradation of atrazine by isolating fungal strains from the environment and investigating their metabolic pathways. In this study, nine fungal strains produced a ligninolytic enzyme. Four fungal strains (*Fusarium* spp. CCLM-GU, CCLM-DF, CCLM-IB, and CCLM-GW) were involved in catabolism. They degraded atrazine into deethylatrazine, whereas the other five fungal strains efficiently degraded atrazine. Moreover, two strains of CCLM-GW and CCLM-GU more effectively reduced toxicity and residual levels according to the *Allium cepa* (onion) assay results. *Aspergillus niger* is a potential filamentous fungal species that degrades pollutants such as pectin, cellulose, dyes, heavy metals, and other organic and inorganic polymers. To degrade atrazine, this fungal strain was isolated from the soil, and its degradation efficiency was characterized by analyzing different parameters like temperature, soil humidity, and co-substrate for bio-stimulation. This research showed that using *O. ficus indica* 10 wt% as a co-substrate with 80% humidity achieved 75% degradation in six days [90]. In another study, a fungus strain, *Pleurotus ostreatus* INCQS 40310, was isolated from a contaminated site, and was capable of degrading atrazine from 90.3% to 94.5% within 10–15 days in the presence of FeSO_4_ and MnSO_4_ salts. Furthermore, different metabolites of atrazine, the purification of laccase enzymes from the fungus strain, and their activity in the degradation process were evaluated. The findings showed that the fungal strain effectively degraded atrazine residues and was considered a potential agent [91].

To gain the best degradation in the natural environment and for wastewater treatment, recently, the fungal strain *Bjerkandera adusta* was isolated from rotten wood surfaces and its degradation efficiency was optimized using various parameters, such as the concentration of atrazine, pH range, and the effect of temperature and biomass. The results of this study revealed that fungal strains survive better at a pH range of 2–8, a temperature of 16–32 °C, biomass of 1–5 g, and a concentration of 25–100 mg/L, and degraded 92% in six days [92]. To degrade toxic residues of propazine in wastewater, the fungal strain *P. ostreatus* INCQS 40310 was cultured in two different media, PMP 7 and PMP 12, and their degradation ability was assessed. The results showed that the fungal strain gave satisfactory results in PMP 12 culture media and degraded 90% of propazine [78]. The fungal species *P. mutilus* was isolated from a contaminated site, and the adsorption of the metribuzin herbicide was examined. This research concluded that the fungal strain adsorbed 70% of metribuzin at an initial concentration of 200 mg/L, a temperature of 25 °C, a pH range of 2–3, biomass content of 3 g, and a rotation speed of 250 rpm [70]. *Funneliformis mosseae* was combined with biochar and the authors investigated its degradation efficiency for decomposing simazine in two agricultural soils. The findings expressed that the mutual relationship between arbuscular mycorrhizal fungi, plants, and biochar-mediated soil significantly inhibited simazine decomposition. In contrast, soil inoculated with arbuscular mycorrhizal fungi and biochar efficiently reduced the concentration of simazine. This study attributed these results to variation in the soil’s sorption capacity due to biochar amendment and the inoculation of fungal strains [93]. A soil yeast was isolated from the soil, identified as *Lipomyces starkeyi*, and its degradation efficiency was tested against s-triazine herbicides. The results of this study showed that the yeast strain efficiently degraded and assimilated triazine herbicides (atrazine, ametryn, cyanazine, prometryn, and simazine) [94]. Recently, to remove atrazine and other toxic compounds (nitrate, magnesium, phosphorous, zinc, oxadiazon, and triallate) from wastewater, a microalgae, Scenedesmus sp., was immobilized in a continuous flow reactor system. The results revealed that at temperatures of 20–35 °C and a one-week hydraulic retention time, the algal beads were able to degrade 97% of nitrate and 70% of atrazine. The capability of algal beads to degrade all pollutants in actual wastewater was also tested. The findings showed that algal beads also perform well and degrade 92%, 100%, 99.9%, and 92% of nitrate, magnesium, phosphorous, and zinc, respectively. However, some herbicides diffused back into the solution toward the end of the treatment process [95].

### 2.1. Degradation of S-Triazines Using Microbial Enzymes and Genes

Microbial species metabolize s-triazine herbicides through enzyme-catalyzed reactions. The hydrolytic remediation of substituents on the s-triazine ring is catalyzed by enzymes from the amidohydrolase superfamily, generating a cyanuric acid metabolite. Further, the hydrolytic reaction converts cyanuric acid into ammonia and carbon dioxide. Thus, the efficiency of single and microbial consortiums in biodegradation mainly depends on various kinds of intracellular and extracellular enzymes and potential genes are produced, which are directly or indirectly involved in the degradation process and enhance the degradation rate of xenobiotics [96]. Recently, Zhou et al. [97] found that purified amidohydrolase enzymes, i.e., N-isopropylammelide isopropylaminohydrolase (AtzC) from bacterial strain *Leucobacter triazinivorans* JW-1, were effective against the biodegradation of s-triazine herbicides. Purified *AtzC* is involved in the catalyzation of the amidohydrolysis of N-isopropylammelide and converted into cyanuric acid. The activity of catalytic residues (His253, His60, His62, His211, Asp307) in *AtzC* during the process of N-isopropylammelide hydrolysis was also reported based on molecular docking and bioinformatic analysis. Jiang et al. [98,99] isolated a bacterial strain, *Arthrobacter* sp. DNS10, from polluted soil. They evaluated bacterial genes and enzymes for the degradation of atrazine in the presence and absence of the nicosulfuron herbicide, which is widely applied in the agricultural sector in combination with s-triazine herbicides. They found a carbohydrolase gene (*trzN*) from *Arthrobacter* sp. DNS10, and revealed that 76% of atrazine was degraded without nicosulfuron in 48 h. At concentrations of 1, 5, and 10 mg/L and in the presence of nicosulfuron, the bacterial genes could degrade 63.9%, 49.1%, and 42.6, respectively.

Liang et al. [62]’s gene knockout study showed that the plant growth-promoting rhizobacteria *Pseudomonas* sp. DY-1 could degrade prometryn using Baeyer–Villiger monooxygenase. The result of gene knockouts demonstrated that enzymes are actively involved in the degradation of prometryn, which provides new molecular insights for removing s-triazine herbicides from the polluted environment. James et al. [100] reported the biodegradation of atrazine by two bacterial strains, *Pseudomonas* spp. strains ACB and TLB, and screened intracellular crude enzymes’ ability to enhance the degradation rate. Their results showed that enzymes purified from the strain ACB revealed the highest degradation of atrazine at 71% at temperatures of 30–40 °C and pH 6, whereas the enzymes extracted from the bacterial strain TLB were able to degrade at 48–46% under the same conditions. Furthermore, the purified enzymes were exposed to protein profiling investigation. The findings of protein profiling demonstrated that atrazine induced the expression of CoA ester lyase and alkyl hydroperoxide reductase in the strains ACB and TLB, respectively.

Wackett et al. [101] proposed that *Pseudomonas* sp. ADP carried a plasmid, pADP-1, which was entirely sequenced and encompasses potential genes involved in atrazine degradation. Furthermore, this study explained that the recombinant *Escherichia coli* strain contains atrazine chlorohydrolase enzymes constructed and chemically cross-linked to produce catalytic particles employed for atrazine degradation in soil. A *Pseudomonas* sp. strain, NRRLB-12227, potentially degraded s-triazine herbicides and was used as a sole source of carbon and energy. Further, the *trzD* genes encoded by cyanuric acid amidohydrolase were cloned into pMMB277, capable of deleting s-triazine herbicides. The *Arthrobacter aurescens* strain TC1 was screened without enrichment by plating atrazine-polluted soil directly onto atrazine-clearing plates. The rapid growth of this strain in a liquid medium in the presence of atrazine at a concentration of 3000 mg/L was observed. The bacterial strain utilized herbicides such as carbon, energy, and nitrogen. Moreover, this bacterial strain provides rapid growth in other s-triazine-related compounds such as ametryn, atratone, cyanazine, prometryn, and simazine. The results of the PCR experiment identified different s-triazine-degrading genes from the *Pseudomonas* sp. strain ADP, which showed a high resemblance to *atzB* and *atzC* but not to *atzA* [102]. Balotra et al. [103] reported that AtzDEF proteins were shown to form an extensive range of enzymes that have the potential to degrade s-triazine metabolites, such as cyanuric acid, and generate non-toxic products such as ammonia and carbon dioxide. *Paenarhtrobacter aurescens* TC1 was originally isolated from atrazine-polluted soil. It converts atrazine into a cyanuric acid metabolite.

Genes (*atzABCDEF*) involved in the degradation of s-triazine herbicides might be placed in a larger plasmid of different microbial species, especially for bacterial chromosomes [104,105]. An indigenous *β-proteobacterium* strain, CDB21, was screened from soil that potentially degraded simazine. Via molecular mechanisms, a complete set of simazine-degrading genes (*atzABCDEF*) were investigated, and their degradation capability examined. The results revealed that significant degradation of simazine was observed. The genes *trzN*, *atzB*, and *atzC* in *Paenarhtrobacter aurescens* TC1 revealed that they are potentially involved in the degradation of the atrazine herbicide. Moreover, the genes *trzN*, *atzB*, and *atzC* are situated on the plasmid of *Paenarhtrobacter aurescens* TC1. This study explained that all the enzymes that potentially degrade atrazine encoded by *trzN*, *atzB*, and *atzC* belong to the amidohydrolase superfamily [106]. In another study, a bacterial strain, *Arthrobacter* MCM B-436, was cultured from vetiver rhizosphere soil and the authors investigated their degradation efficiency by free cells and using their associated genes. The results revealed that the atrazine degradation genes *atzB*, atzC, *atzD*, and *trzN* of *Arthrobacter* MCM B-436 and their enzymes were all located in its genome and efficiently degraded atrazine into its derivatives [59]. The role in atrazine degradation of *trzN* was confirmed in *Arthrobacter* sp. C3. Triazine hydrolase encoded by *trzN* can degrade atrazine into a nontoxic hydroxyatrazine [40]. *trzN*, *atzB*, and *atzC* in *Arthrobacter* sp. SD3-25 was revealed to degrade simazine into cyanuric acid. Interestingly, the efficiency of degrading simazine can be increased with increasing sucrose concentration [107]. Fajardo et al. [108] isolated four different bacterial species from polluted soil, recognizing them as *Acinetobacter lwoffii*, *Pseudomonas putida*, *Rhizobium* sp., and *Pseudomonas* sp. Through fluorescence, in situ hybridization, and qPCR investigation, the target gene *atz* was identified to potentially degrade simazine. The results revealed that *Rhizobium* sp. presented the maximum removal of simazine at 71.2% and mineralization was 38.7%, while the lowest degradation and mineralization were observed in *A. lwoffii* at 50.4% and 22.4%, respectively. However, this proposed study and molecular insights on the basis of s-triazine-degrading genes could be considered future tools for the degradation of xenobiotics and other organic and inorganic pollutants.

### 2.2. Microbial Metabolic Pathways of S-Triazine Herbicides

The conversion of s-triazine herbicides into their derivatives leads to the same or more toxicity than their parent compounds [109]. The microbial metabolism of agrochemicals helps alter the toxicity of pesticides via microbial metabolic pathways, which leads them to become wholly degraded into less toxic compounds [110]. Moreover, a complete understanding of s-triazine herbicide biodegradation pathways would help to better utilize these beneficial microbes for more sustainable remediation [111].

A native bacterial strain was isolated from soil and recognized as *Leucobacter* sp. based on its morphological appearance, physiological appearance, and 16S rDNA gene sequence. The degradation efficiency of the strain *Leucobacter* sp. JW-1 was tested again with different s-triazine herbicides. The results revealed that this strain completely degraded ametryn and prometryn, whereas 99% simetryn, 41% propazine, 43% atrazine, 28% simazine, 12% terbuthylazine, 10% prometon, and 13% atraton were degraded, at a concentration of 50 mg/L in two days. Moreover, the bacterial degradation mechanism of prometryn was also investigated, which degraded into 2-hydroxypropazine and methanethiol through a new hydrolysis pathway. Further, 2-hydroxypropazine was converted into cyanuric acid via two sequential deamination reactions [53] (Figure 4). To degrade ametryn herbicide into less toxic metabolites, an entomopathogenic fungus, *Metarhizium brunneum*, was utilized and its degradation efficiency investigated. The results revealed that the fungal strain could tolerate a 100 mg/L concentration of ametryn with the addition of 2% glucose. The results of liquid chromatography and mass spectrometry analysis revealed that after a seventeen-day culture of the fungal strain in mineral salt medium, ametryn was degraded into four metabolites: 2-hydroxy atrazine, ethyl hydroxylated ametryn, S-demethylated ametryn, and deethylametryn [112].

For the biodegradation of the s-triazine herbicide terbuthylazine, a mixed bacterial culture, M3-T, was isolated from soil and its degradation efficiency examined in a mineral salt medium and agricultural soil. The results revealed that M3-T efficiently degraded terbuthylazine in a mineral salt medium after three days of culture. Moreover, liquid chromatography and mass spectrometry analysis revealed that terbuthylazine was broken down into five metabolites and categorized into major and minor metabolites. The primary metabolites were converted into hydroxyterbuthylazine and N-t-butylammelide, and the minor metabolites were desethylterbuthylazine, hydroxydesethylterbuthylazine, and cyanuric acid [113]. The capability of soil microbial species to degrade propazine residues in sterilized and unsterilized soil was investigated. The results revealed that after 7–11 days, the soil microbial species degraded 90–94% propazine, respectively, at a concentration of 10 mg/kg. An investigation of the transcript level of the degradative genes *AtzB*, *AtzC,* and TrzN revealed that these genes were promoted by propazine and played crucial roles in the degradation mechanism, whereas the results of ultra-performance liquid chromatography and mass spectrometry analysis revealed that propazine breaks down into five significant intermediates: hydroxyl-, methylated-, dimericpropazine, ammeline, and ammelide [114].

A native bacterial strain, FJ1117YT, was screened from a contaminated site through an enrichment culture for the effective bioremediation of simetryn. The consortium was proficient in removing the simetryn herbicide as the sole source of carbon, sulfur, and energy in a mineral salt medium. The strain FJ1117YT was recognized as *Rhodococcus* sp. based on its morphological and physiological properties and the 16S rRNA gene sequence. Analysis of liquid chromatography–mass spectrometry and nuclear magnetic resonance revealed that simetryn was degraded by separating the methylsulfinyl analogue as the primary metabolite and identifying the methylsulfonyl intermediate and the hydroxy analogue as secondary metabolites. The findings showed that the methylthio group was rapidly oxidized and hydrolyzed via bacterial strain. Additionally, this bacterial strain can effectively metabolize other s-triazines herbicides such as ametryn, desmetryn, dimethametryn, and prometryn by comparable degradation pathways [56]. The bacterial strain *Pseudomonas* sp. ADP is considered a model strain that potentially degraded s-triazine herbicides from the environment. This bacterial strain used especially atrazine and other s-triazine herbicides as a carbon and energy source. The enzymes purified from bacterial strain efficiently converted atrazine into cyanuric acid are encoded by the *atzA*, *atzB,* and *atzC* genes, while cyanuric acid is further divided by *atzDEF* genes. The herbicide simazine is also degraded by these catabolic enzymes and genes [115].

A new bacterial strain for effective atrazine remediation was isolated from soil by enrichment culture technique and identified as *Klebsiella variicola* FH-1 based on physiological, morphological, and 16s rDNA sequences. This bacterial strain rapidly grew in a mineral salt medium and utilized atrazine as a sole source of carbon and energy. The optimum degradation efficiency of FH-1 was investigated by various parameters such as temperature, pH, initial concentration of atrazine, and the effect of additional sources in the form of metals such as zinc and copper. Results of the degradation experiment revealed that this bacterial strain in mineral salt medium efficiently degraded 81.5% of atrazine at a temperature of 25 °C, pH 9, and initial concentration of 50 mg/L in 11 days. Among metal ions, due to the utilization of 0.2 mM of zinc, the degradation rate was significantly accelerated by 72.5%, while copper reduced the growth of bacterial strain and decreased the degradation of atrazine. While the results of high-performance liquid chromatography revealed that bacterial strain breakdown atrazine into 2-hydroxyl-4-ethylamino-6-isopropylamino-1,3,5-triazine, 2-hydroxyl-4,6-bis(ethylamino)-1,3,5-triazine and 4,6-bis(ethylamino)-1,3,5-triazin-2(1H)-one. Additionally, PCR analysis revealed that three degrading genes *atzC*, *trzN*, and *trzD* were identified, which were potentially involved in the enzymatic degradation of atrazine [116]. In another study *Arthrobacter* sp. ZXY-2, *Aspergillus niger* Y3, *Agrobacterium* sp. WL-1 and make a composite that potentially degraded atrazine herbicide in a mineral salt medium. The composite was immobilized in biochar and made pellets which 61% more degraded atrazine as compared to free cells. Ultra-performance liquid chromatography and mass spectrometry analysis revealed that atrazine was divided into metabolites such as hygdoxyatrazine, desethylatrazine, deisopropylatrazine, deisopropyldeethylatrazine, hydroxytrazine, deisopropylhydroxyatrazine, and desethylhydroxyatrazine. The complete degradation pathway is presented in a previous study [117]. Atrazine is highly intractable to biodegradation compared to cyanazine and under methanogenic conditions. The initial disappearance of chemicals may be due to nonbiological processes, and the involvement of biological processes can be achieved through the enrichment culture technique. Degradation of cyanazine and dicamba was observed using this technique, and the degradation intermediates were cyanuric acid and 3,6-dichlorosalicyclic acid, respectively. The initial reaction of dicamba degradation is O-demethylation under methanogenic conditions [118,119].

## 3. Phytoremediation of S-Triazine Herbicides

The biodegradation of xenobiotics and other emerging pollutants from the natural environment is an emerging issue. Three general methods are extensively applied: indigenous microbial species, bioaugmentation, and phytoremediation [120,121]. The phytoremediation of pollutants is considered adequate, low-cost, environmentally friendly, and more efficient for the degradation of contaminants present in the air, soil, and water [122]. Plants can potentially degrade pesticides, including s-triazine herbicides, in various contaminated locations [123]. Microbial communities placed in the root zone of vegetation provide potential genes and play a pivotal role in biodegradation [124]. The degradation process of pesticides in the soil is a symbiotic reaction between microbial plant species, which enhances soil nutrients [125]. Moreover, vegetation provides a suitable environment for microbial communities that degrade pesticides and enhance the growth of plants [126]. Different plant species are presented in Table 2 that effectively remediate toxic s-triazine residues from soil and water. The phytoremediation of pollutants is divided into four categories: (i) the direct uptake and accumulation of pesticides and subsequent metabolism in plant tissues, an efficient process of pesticide degradation; (ii) the transpiration of volatile organic hydrocarbons by leaves; (iii) the release of exudates that stimulate microbial activity and biochemical transformations in the soil; and (iv) the improvement of mineralization at the root–soil interface by microorganisms [127]. Transgenic plants are very beneficial for the phytoremediation of pesticides. A P450 isoenzyme, CYP1A2, was purified for the remediation of simazine from the Arabidopsis thaliana transgenic plant. The results revealed that CYP1A2 could be used as a potential agent for the mineralization of simazine and converted into less toxic substances, which induced further uptake and enhance plant growth. Moreover, this study further indicated that the application of simazine at a concentration of 250 µmol significantly increased plant growth, while for non-transgenic plants, a concentration of more than 50 µmol damaged the plant and burned it [128]. In another study on the degradation of atrazine, transgenic rice (*Oryza sativa*) was considered a superior applicant. The results revealed that rice contains cytochrome P450 monooxygenase CYP1A1, CYP2B6, and CYP2C19, which are encoded by degrading genes and metabolize atrazine into less toxic compounds [129]. To degrade prometryn herbicide through a phytoremediation mechanism, vetiver grass (*Chrysopogon zizanioides* L.) was investigated. This grass was grown in a hydroponic medium and placed in a greenhouse with prometryn and proper control. The results revealed that vetiver grass potentially degraded the prometryn herbicide and its half-life was noted as 11.5 days. The first-order remediation kinetics was (Ct = 1.8070e − 0.0601t) [130]. In another study on the remediation of prometryn, a greenhouse investigation was conducted, and *Canna indica* plants were used. The results revealed that this plant provided satisfactory growth at a concentration of 0.5 mg/L and degraded prometryn more efficiently than the control. The half-life of prometryn was reduced in this study by up to 17 days, while the 99% dissipation in the control was 109 days. The combined analysis of prometryn in the solution and plant uptake revealed that it was completely removed by *Canna indica* within 10–16 days [131].

The proficiency of two types of grass, tall fescue (*Festuca arundinacea*) and orchard grass (*Dactylis glomerata*), for the remediation of terbuthylazine was tested in hydroponic media with various concentrations from low to high: 0.06, 0.31, 0.62, and 1.24 mg/L. The results revealed that the grass tall fescue exhibits outstanding growth in hydroponic media and efficiently degrades terbuthylazine. Moreover, this study identified that two enzymes, glutathione S-transferase and ascorbate peroxidase, play a pivotal role in detoxification, and grass was not damaged due to the higher application of herbicide. Meanwhile, adverse effects of terbuthylazine on orchard grass were observed [132]. For the phytoremediation of ametryn herbicide from polluted soil, four different plants (wheat, maize, ryegrass, and alfalfa) were selected and examined for their capability. Besides this, the accumulation percentage of ametryn in various parts of plants, soil enzyme activity, the efficiency of antioxidant and degradation enzymes, and low-molecular-weight organic acids were also investigated. The results revealed that the highest concentration of ametryn was observed in the shoots and roots of wheat and alfalfa, whereas the accumulation percentage was much lower in ryegrass. However, ryegrass showed higher tolerance against the ametryn herbicide and was degraded in polluted soil by plant tissues and enzymes [133]. Metribuzin degradation from soil was achieved at a 5 mg/kg concentration through the amendment of municipal waste compost and sheep manure. A laboratory bioassay was carried out, and the phytotoxicity of metribuzin was investigated on oilseed rape (*Brassica napus* L.). The results of this study revealed that the soil with the amendment of municipal waste compost degraded 88.8% of metribuzin in oilseed rape plants, and sheep manure degraded 72.2%. Moreover, the soil without any amendment of manure for the oilseed rape plant was able to degrade 59.8% of metribuzin. This study concluded that municipal carbon manure was the superior agent enhancing metribuzin degradation in oilseed rape soil [134]. Plants degrade atrazine through three metabolic pathways. The superior pathway of atrazine remediation in some resistant weeds is glutathione conjugation, in which glutathione s-transferase displaces the chlorine atom on the 2-carbon atom of atrazine [135]. The second mechanism is hydrolysis, where the chlorine atom in atrazine is substituted with a hydroxyl group. The third pathway is N-dealkylation, in which cytochrome P450 monooxygenases remove the ethylamino and isopropyl amino side chains [136]. An investigation of the phytoremediation of atrazine from polluted soil by *Pennisetum alopecuroides* (L.) was conducted. The results revealed that *Pennisetum alopecuroides* (L.) enhanced the degradation of atrazine by 15.22 to 51.46%, and only a small concentration of atrazine was observed in the roots and shoots of plants [137]. Recently, Pérez et al. [138] studied the degradation of atrazine by *Typha latifolia* and monitored the physical appearance of plants and the uptake and translocation of atrazine by various parts of plants, and bioaccumulation and removal by selected plants was also investigated. The results revealed that plant tissues degraded atrazine into its derivatives (esethylatrazine and desisopropylatrazine). The detoxification of atrazine was regulated by *Typha latifolia* via different metabolic pathways. Moreover, the plant study showed that the plants’ weights were reduced after 21 days, and their transpiration rates declined after 28 days. In another study, atrazine was degraded into diaminochlorotriazine and hydroxyatrazine end-products by employing two plant species. This study revealed that both plants rapidly degraded atrazine compared to the control [139]. Recently, Yu et al. [140] studied the degradation of three triazine herbicides (terbuthylazine, ametryn, and atrazine) in maize plants and examined the effect of salicylic acid, which is a natural plant growth regulator, on the detoxification of three herbicides. The results revealed that by applying 6 mg/kg of triazine herbicides, the growth and chlorophyll of maize plants significantly declined, while the membrane permeability was enhanced. By applying salicylic acid at a 5 mg/kg concentration, the triazine herbicides increased phytotoxicity and chlorophyll content and decreased cellular damage in maize plants.

**Table 2 microorganisms-11-02558-t002:** Phytoremediation of s-triazine herbicides.

Target S-Triazine Herbicide	Name of Plant Species	Statement	References
Atrazine	*Potamogeton crispus* *Myriophyllum spicatum*	Both species efficiently degraded atrazine up to 90%, and the half-life of atrazine was recorded at 8.60 and 9.72 days, respectively, for both plants.	[141]
	*Lolium perenne* *Festuca arundinacea* *Hordeum vulgare* *Zea mays*	All plant species were able to degrade atrazine into their metabolites. The final degradation by plant species was 88.6–99.6%, while in unplanted plots, the degradation of atrazine was 63.1–78.2%.	[142]
	*Panicum virgatum*	The results revealed that this plant species detoxifies atrazine into its non-toxic metabolites, which are present in the leaf only.	[143]
	*Lolium perenne* L.	In this investigation, the electrokinetic-assisted phytoremediation of atrazine was tested. The results revealed that plant species degraded atrazine 27% faster as compared to natural attenuation.	[144]
	*Phaseolus vulgaris* L.	Through the combination of rhizome microbial species and plants, 76.63% of atrazine was degraded at a concentration of 50 mg/L in soil.	[145]
	*Eremanthus crotonoides* *Inga striata*	Atrazine was degraded by both plant species, and these plants prohibited the leaching of atrazine residues into the underground water system.	[146]
	*Pistia stratiotes* *Eichhornia crassipes*	Atrazine was spiked in both plants at a concentration of 10–100 µg/L in 5 L pots. The results demonstrated that atrazine residual concentration was less in the plant system as compared to the water solution.	[147]
	*Potamogeton crispus* *Myriophyllum spicatum*	Both plants absorbed atrazine and converted it into their metabolites. Additionally, chromatography analysis revealed that an equal concentration of atrazine was adsorbed by roots and shoots.	[148]
Ametryn	*Ananas comosus*	Gas chromatography and mass spectrometry analysis revealed that the residues of ametryn were degraded effectively, and 0.002 mg/kg of atrazine in the plant and 0.005 mg/kg in the soil were detected.	[149]
	*Digitaria horizontalis*	The results revealed that this plant was able to treat ametryn in avocado fields up to 6 L/ha.	[150]
Cyanazine	*Panicum dichotomiflorum* *Setaria viridis* L. *Zea mays*	The results revealed that cyanazine more rapidly metabolizes in Zea mays and a large amount of cyanazine was absorbed by the root system as compared to other plant systems.	[151]
	*Sorghum bicolor* L.	In this investigation, cyanazine and atrazine were applied and the degradation efficiency of the plant was tested. The results showed that the half-life of cyanazine was 15 days, and in atrazine, 60 days was observed.	[152]
	*Zea mays*	The breakdown of cyanazine in corn was investigated and the results revealed that it converts into hydrolysis products such as amide and hydroxy acid.	[153]
	*Bromus inermis Leyss* *Elymus junceus Ficsch.* *Agropyron intermedium*	All the grasses completely removed cyanazine, and a very small concentration of 0.02 mg/kg was detected via gas chromatography.	[154]
	*Typha latifolia*	This plant was investigated for the removal of s-triazine herbicides (atrazine, ametryn, and dimethametryn). The results revealed that the flower of this plant effectively removed residues of all treated pesticides and this method is cost-effective, easy, and ecofriendly.	[155]
Metribuzin	*Glycine max* L.	The results revealed that the tetraploid of this plant showed great resistance, while diploid plants are highly susceptible. Moreover, tetraploid plants efficiently degraded metribuzin into non-toxic substances.	[156]
	*Triticum turgidum* L.	The residual concentration of metribuzin was tested via gas chromatography. The results showed that a very small quantity of metribuzin is present in plants, and others are degraded into less toxic substances that are further used by the plant system.	[157]
	*Glycine max* L.	The results demonstrated that metribuzin was absorbed by the plant system efficiently. About 97% of metribuzin was absorbed by the root system and transformed into its metabolites.	[158]
	*Daucus carota* L.	The residues of metribuzin and other herbicides were monitored in this plant. The results indicated that the highest uptake was observed by the root system as compared to the leaves and shoots.	[159]
	*Bromus tectorum* *Triticum aestivum*	The tolerance rate of metribuzin was investigated by both plant species. The results indicated that *Bromus tectorum* absorbs and transforms a two-times-higher concentration of metribuzin as compared to other plants.	[160]
Prometryn	*Phaseolus vulgaris* L.	A 10–100 µM concentration of prometryn was applied. The results indicated that the bean plant tolerated prometryn and was a reliable material that could be used for herbicide pollution sensing.	[161]
	*Arachis hypogaea*	Various degradation percentages of prometryn were observed in different parts of plants. The results of gas chromatography revealed that the lowest concentration of metribuzin detected was 0.02 mg/kg.	[162]
	*Angelica acutiloba*	The residues of prometryn were degraded more effectively by the root system and the final concentration detected was 0.0368 in dried roots. Meanwhile, after harvesting of the plant, a concentration of prometryn of 0.0464 was observed via gas chromatography and mass spectrometry analysis.	[163]
	*Pontederia cordata* L. *Typha latifolia* L. and *Cyperus alternifolius* L.	These three plants degraded s-triazine herbicides (atrazine, prometryn, and simazine) in more than 25% of the wastewater.	[164]
Propazine	*Zea mays* *Triticum aestivum* *Brassica napus*	The results indicated that propazine was effectively degraded by all plants. Six metabolites of propazine were detected in the wheat plant. Through the foliar application of salicylic acid (plant growth regulator), the herbicide rapidly degraded and the growth of the plant increased.	[165]
	*Gossypium hirsutum*	The results revealed that a higher concentration of propazine is tolerated by this plant and causes less injury when applied at the pre-emergence stage. This plant uptakes the residue of propazine rapidly.	[166]
	*Zea mays*	The results revealed that 96% of propazine was degraded at an initial concentration of 50 mg/L by maize and bacteria-immobilized beads.	[77]
Simazine	*Typha latifolia*	The results revealed that after seven days, the residues of 65%simazine were degraded. This study concluded that this plant is an excellent candidate for the removal of simazine from polluted soil.	[167]
	*Oryza sativa*	The human cytochrome P450 CYP1A1 gene was injected into rice plants. The results concluded that human genes make it possible to tolerate simazine and atrazine herbicides and rapidly degrade their residues in culture media and polluted soil.	[168]
	*Canna hybrida*	The results revealed that 80% of simazine was degraded after seven days by plants and considered a good candidate for the removal of herbicides.	[169]
	*Zea mays*	In corn plants, the absorption and translocation of atrazine and simazine were studied. The results revealed that both herbicides were metabolized into non-toxic products.	[170]
	*Zea mays*	The results indicated that simazine was rapidly degraded by the root and shoot of corn plants in polluted soil.	[171]
Terbutryn	*Myriophyllum aquaticum*	The uptake translocation and metabolism of atrazine and terbutryn were investigated. The results revealed that a lower concentration of terbutryn was observed in roots and shoots as compared to atrazine. Moreover, plants metabolized both herbicides into less toxic substances.	[172]
	*Gammarus fossarum* *Asellus aquaticus*	The results indicated that complete degradation of terbutryn was observed in both plant species, and a half-life of terbutryn of 5 h was recorded.	[173]
	*Gossypium hirsutum* *Phaseolus vulgaris*	Terbutryn was degraded by both plant species and a higher concentration was mineralized by the root system of a plant.	[174]
Terbuthylazine	*Lolium multiflorum*	An experiment was performed using *Lolium multiflorum* and terbuthylazine at aqueous concentrations of 0.5, 1.0, and 2.0 mg/L. The plant was able to remove 38%, 42%, and 33% of the initial concentrations, respectively, in 240 h; over this period, the capacity of plants to absorb the herbicide at 0.5 mg L1 increased from 1.58 to 3.50 mg g1 of fresh weight.	[175]
	*Lolium multiflorum* L.	The ryegrass efficiently degraded terbuthylazine through the activation of glutathione s-transferase and peroxidase enzymes.	[176]
	*Lemna minor*	This study concluded that duckweed is suitable for cleaning terbuthylazine-polluted water from the environment and degrading toxic residues through the activation of peroxidase and catalase enzymes.	[177]
	*Typha latifolia* L.	The plant degraded terbuthylazine into its metabolites in the wetland system. These results concluded that this plant could be used for the remediation of pollutants from wastewater.	[178]
	*Typha latifolia* L.	This study concluded that the degradation of terbuthylazine showed gradient behavior via the depth of the sediment substrate of wetlands, and its metabolites followed the effect of the biotic and abiotic mechanisms of degradation in the bioreactor substrate.	[179]

Phytodegradation summary: This table provides information about specific plant species that are effective in removing or degrading s-triazine herbicides from the soil or water. Phytoremediation is an environmentally friendly approach to remediating contaminated sites using plants. This table also includes details about target compounds and their success rates in the phytoremediation process.

## 4. Nano-Remediation of S-Triazine Herbicides

The application of nanotechnology in various fields, such as industries, health and medicine, agriculture, environmental issues, food technology, biotechnology, etc., is attracting great interest due to its many advantages [180]. To solve severe environmental and agricultural problems, an extensive range of nanoparticles are synthesized via biological, physical, and chemical methods (Figure 4). Nanomaterials for cleaning up the environment and the removal of various kinds of pollutants in soil, water, and sediments are applied on a laboratory scale and in natural environments and have provided satisfactory results [181]. These nanoparticles are designed with a 1–100 nm dimension [182]. Due to their smaller size, many nanoparticles are equally spread into miniature crevices in the subsurface and stay suspended in groundwater for an extended period [183]. However, due to the extensive range of practical applications of nanoparticles and nanomaterials, they have been considered an efficient agent for removing a wide range of pollutants from soil and water in the real environment, which are impossible to biodegrade [184]. Different nanomaterials and their optimum conditions for removing s-triazine herbicides, especially from polluted water, have been discussed in Table 3.

Ali et al. [185] investigated the absorption of ametryn herbicide using iron nanoparticles as adsorbents. To optimize the degradation of ametryn, a batch experiment was carried out at 30 µg/L, with a contact time of thirty minutes and an adsorbent nanoparticle concentration of 2.5 g/L at a temperature of 20 °C. The residual concentration of ametryn was investigated via high-performance liquid chromatography, and the results revealed that ametryn was significantly removed from wastewater. This study indicated that the sorption method was rapidly performed and considered eco-friendly and lower in cost, and effectively degraded contaminants from water bodies in real environmental conditions. Recently, Cheng et al. [186] studied the degradation of four s-triazine herbicides, atrazine, ametryn, metribuzin, and prometryn, on two typical natural zeolites (clinoptilolite and Fe-mordenite) and two common clays (kaolin and attapulgite) with microporosity in batch experiments. The results indicated that all these s-triazine herbicides highly depend on pH and mineral materials. Moreover, iron zeolites at a concentration of 0.05 mmol/L efficiently increased the sorption process of atrazine, ametryn, and prometryn by enhancing the protonation of s-triazine molecules in the interfacial region between the mineral surface and bulk solution. Atrazine was degraded via the adsorption process using iron nanoparticles. The composite of iron nanoparticles was synthesized, analyzed, and employed for the remediation of atrazine in water. The residual concentration was analyzed via gas chromatography and mass spectrometry. The results indicated that at a temperature of 20 °C, pH 7, and contact time of 30 min, the highest removal of atrazine 95 was recorded at a concentration of 2.5 g/L. This study concluded that this adsorption method is efficient, inexpensive, and rapidly performed for the degradation of atrazine in water samples [187].

Two mesoporous metal oxide-based nanomaterials, aluminum oxide and iron oxide, were synthesized and investigated for the remediation of simazine in an aquatic system. The removal rate was optimized using various parameters such as temperature, pH, contact time, simazine concentration, and sorbent dose. The results revealed that the highest sorption of simazine was recorded at pH 6.5 for aluminum oxide and 3.5 for iron oxide. Moreover, the sorption rate of simazine was higher using iron oxide than aluminum oxide nanocomposites [188].

The removal of simazine was tested using zinc oxide/graphene oxide nanocatalysts under a batch experiment. At pH 2, 30 mmol of nanocatalysts performed efficiently, and a higher concentration of simazine was removed compared to pure zinc oxide, with the removal rate 62% higher than the rate of photolysis [189]. In another study for the removal of atrazine, a novel photocatalyst type of titanium oxide nanoparticle was manufactured using waste material. The remediation study evaluated different parameters, such as the concentration of atrazine, contact time, and catalyst dosages. The results revealed that using a titanium oxide nanocatalyst for the photodegradation of atrazine was an efficient, lower-cost, eco-friendly, simple, and faster technique that can be applied to remove other pollutants [190]. The degradation of the secbumeton s-triazine herbicide from water using a nano-adsorbent was evaluated. The residual concentration was determined via gas chromatography and mass spectrometry. The findings indicated that at an initial concentration of 30 µg/L of secbumeton, pH 7, a temperature of 20 °C, a dose of nano-adsorbent of 2.5 g/L, and a contact time of 30 min, the nano-adsorbent was able to degrade 90% of the herbicide in water. This study showed that this method was very economical and can be efficiently applied to remove contaminants from water on a large scale [191]. For the removal of ametryn from water, natural zeolite nanoparticles were synthesized. The properties of the nanoparticles were evaluated, and a Box–Behnken response surface methodology was applied to investigate all parameters affecting the adsorption mechanism continuously. The results demonstrated that at a temperature of 43 °C, pH 6, 2 g of adsorbent, and a contact time of 2 h, the nanoparticles were able to degrade 64.12% of ametryn in the water samples [192].

**Table 3 microorganisms-11-02558-t003:** Activation of nanoparticles and their important roles in the removal of s-triazine herbicides.

S-Triazine Herbicide	Type of Nanoparticles	Function	References
Atrazine	Fe_3_O_4_	At optimum conditions, nanoparticles performed well and efficiently degraded 80% of atrazine in an aqueous solution with a short retention time of 20 min. Moreover, atrazine degraded into various intermediate products and finally degraded into less toxic substances.	[193]
	TiO_2_ modified with (Au, Cu, and Ni)	At an initial concentration of 25 mg/L, titanium nanoparticles were able to degrade 48% of atrazine, while their modification with a gold catalyst performed well and degraded 60% of atrazine at a retention time of 300 min.	[194]
	CoFe_2_O_4_	To remove atrazine from wastewater, hybrid cobalt ferrite nanoparticles were synthesized and employed as a highly efficient peroxymonosulfate activator, which rapidly degraded 98.2% of atrazine.	[195]
	TiO_2_ modified with Fe^+3^	At an initial concentration of 25 mg/L, modified titanium oxide performs well at pH 11, and following UV radiation, 99% degradation of atrazine was achieved in water samples.	[196]
	Carbon nanotubes	Carbon nanotubes with three different catalysts (pure metallic pallidum, oxide, and silver-coated pallidum) completely removed atrazine from water.	[197]
	Bi_2_O_3_	At a neutral pH, initial concentrations of 5 mg/L and 1 g/L of Bi_2_O_3_ nanoparticles with an average size of 20 nm were able to degrade 92.1% of atrazine within one hour.	[198]
	Cu–Cu_2_O	Cu–Cu_2_O nanoparticles were optimized and synthesized via a simple route. The results revealed that complete degradation of atrazine was achieved with 30 min of visible light in water samples.	[199]
	Fe_3_O_4_	Mesoporous Fe_3_O_4_ nanoparticles were synthesized for the removal of atrazine from water. The results revealed that they showed higher catalytic ozonation activity and their activation level increased by increasing pH, which effectively degraded atrazine from the aqueous solution.	[200]
	Co_3_O_4_/TiO_2_	Co_3_O_4_/TiO_2_ nanoparticles were produced through the simple sol-gel method. Modified nanoparticles perform catalytic reactions, which effectively degrade atrazine from water samples. Moreover, sulfate radicals play a crucial role in the remediation process.	[201]
	GO-α-γ-Fe_2_O_3_	Graphene oxide and iron oxide nanoparticles were synthesized for the remediation of atrazine from water. Via an adsorption process, atrazine was rapidly degraded.	[202]
Ametryn	Fe	Spherical green iron nanoparticles with a size of 20–70 nm were synthesized by eucalyptus leaf extracts. The results revealed that at a pH range of 2–5 and retention time of 30–240 min, ametryn was effectively degraded from water.	[203]
	Fe_3_O_4_/rGO	The results revealed that at pH 5 and a temperature of 25 °C, using a nanocomposite efficiently degraded 93.61 of prometryn within 70 min in water samples.	[204]
	Fe	In this study, iron nanoparticles at a size of 75 nm were synthesized from the leaves of Tectona Grandis and the authors investigated their degradation efficiency in the water. The results revealed that at optimum conditions, complete degradation was achieved with a retention time of 135 min.	[205]
	TiO_2_	At an initial concentration of 0.4 g/L with a TiO_2_ catalyst, complete removal of ametryn was observed within one hour. Ametryn was further degraded into its metabolites in water samples.	[206]
	GO-Fe_3_O_4_	Reduced graphene oxide sheets were investigated using dopamine and decorated with magnetic Fe_3_O_4_ nanoparticles with an average size of 12 nm via a simple co-precipitation method that produced artificial nano-enzymes. The results revealed that nano-enzymes were able to degrade 92% of ametryn.	[207]
Cyanazine	Fe	An eco-friendly, less costly, and feasible iron nanoparticle was synthesized for the removal of cyanazine from water. The results revealed that nanoparticles at pH 7, a temperature of 25 °C, and a retention time of 30 min performed efficiently and removed cyanazine.	[208]
	TiO_2_	The degradation of simazine in polluted water by TiO_2_ stimulated with solar light was investigated. The results demonstrated that cyanazine and other pesticides rapidly degraded in an aqueous solution.	[209]
	TiO_2_	For the treatment of organic wastewater, an aggression experiment with TiO_2_ nanoparticles was carried out. The results indicated that after 12 h, using particles with a size of 84 nm, residues of cyanazine and other organic compounds gradually decreased.	[210]
Metribuzin	Zero-valent iron nanoparticles	The results demonstrated that zero-valent iron nanoparticles depended on pH, and the degradation of metribuzin was also affected by changing the value of pH. At pH 10, 7, and 4, the degradation of 93.22%, 83.74%, and 70.09% of metribuzin was achieved, respectively, in water samples.	[211]
	Ag-ZnO	The results indicated that upon using 100 mg Ag/ZnO composites, 90% degradation of metribuzin at a concentration of 14 mg/L in 90 min was recorded.	[212]
	SnS_2_	The results showed that metribuzin was degraded into a deaminometribuzin metabolite by attacking SnS_2_ nanoparticles. This study concluded that the degradation of metribuzin in water was very slow using this type of nanoparticle.	[213]
	H-Ag-BMO/TiO_2_	This study concluded that under visible light irradiation, H-Ag-BMO/TiO_2_ shows excellent photodegradation of 93.7% for metribuzin.	[214]
	TiO_2_	The mineralization and degradation of metribuzin using titanium oxide nanoparticles was investigated. The results showed that at a concentration of 100 mg/L of metribuzin and 10 mg/L of titanium oxide nanoparticles, 80% degradation was achieved within 300 min of irradiation.	[215]
Prometryn	Pt-TiO_2_/(Er^3+^:Y_3_Al_5_O_12_@Ta_2_O_5_)	The results indicated that Z-scheme P-TET/(OB or Syn)-HAP sonocatalysts were synthesized. To investigate their sonocatalytic efficiency, various parameters such as ultrasonic irradiation time, inorganic oxidants, used times, and trapping agents on the sonocatalytic degradation of prometryn were studied. The best degradation ratio (80.31% based on N computing and 85.07% based on S atom computing) of prometryn could be achieved for 10 mmol/L K_2_S_2_O_8_, 1 g/L P-TET/OB-HAP sonocatalyst and 150 min ultrasonic irradiation.	[216]
	Al_2_Si_2_O_5_(OH)_4_-H_2_SO_4_	A halloysite was modified with H_2_SO_4_ for the removal of prometryn from aqueous samples. The results demonstrated that modified halloysite performs excellently at pH 5 and degraded 96% of prometryn.	[217]
	Fe_3_O_4_-TiO_2_/rGO	The results indicated that at optimum conditions (nanoparticle catalysts 0.5 g/L, concentration of herbicide 15 mg/L, and pH 5), 94% degradation of prometryn was achieved in water samples.	[218]
	TiO_2_	Hydrogen peroxide was immobilized with titanium oxide nanoparticles, and their degradation efficiency for prometryn in an aqueous solution was investigated. The results indicated that both herbicides were degraded, and the final product, cyanuric acid, was obtained.	[219]
	Fe_3_O_4_/rGO	The results revealed that at pH 5 and a temperature of 25 °C, using nanocomposite efficiently degraded 91.34 of prometryn within 70 min in water samples.	[204]
Prometon	TiO_2_	The photocatalytic remediation of prometon and other s-triazine herbicides was investigated using titanium oxide nanoparticles under simulated solar light. The results showed that all herbicides were efficiently removed from water samples.	[220]
	TiO_2_	Hydrogen peroxide was immobilized with titanium oxide nanoparticles, and their degradation efficiency for prometryn and prometon in an aqueous solution was investigated. The results indicated that both herbicides were degraded, and the final product, cyanuric acid, was obtained.	[219]
Propazine	FeO	The adsorption of propazine in an aqueous solution using iron oxide nanoparticles with modification of carbon nanoparticles was investigated. The results indicated that the modified nanoparticles performed efficiently at low pH.	[221]
	TiO_2_	The degradation of simazine in polluted water by TiO_2_ stimulated with solar light was investigated. The results demonstrated that propazine and other s-triazine herbicides rapidly degraded in an aqueous solution.	[209]
Simazine	TiO_2_-Cu	TiO_2_-Cu nanoparticles were synthesized by anodic oxidation method for the removal of simazine. Findings showed that at optimum conditions 64% photodegradation of simazine was achieved.	[222]
	Fe_3_O_4_/rGO	Results revealed that at pH 5 and temperature 25 °C using nanocomposite efficiently degraded 88.55 of simazine within 70 min in water samples.	[204]
	Au–TiO_2_	The sonophotocatalytic removal of simazine based on Au–TiO_2_ was investigated. Simazine was degraded into its intermediate products and finally mineralized into less toxic substances.	[223]
	Fe_3_O_4_-TiO_2_/rGO	The results indicated that at optimum conditions (nanoparticle catalysts 0.5, g/L, concentration of herbicide 15 mg/L, and pH 5) 90% degradation of simazine was achieved in water samples.	[218]
	TiO_2_	Titanium oxide nanotubes efficiently degraded simazine and converted it into the cyanuric acid final product under UV light.	[222]
	GO-Fe_3_O_4_	The findings indicated that the highest removal of simazine, 97%, was achieved in 50 min of sunlight irradiation at optimum conditions (catalyst loading 0.3 g/L, concentration of simazine 0.3 mM, and pH 5).	[207]
Simeton	Fe_3_O_4_/rGO	The results revealed that at pH 5 and a temperature of 25 °C, using a nanocomposite efficiently degraded 81.22 of simeton within 70 min in water samples.	[201]
	Fe_3_O_4_-TiO_2_/rGO	The results indicated that at optimum conditions (nanoparticle catalysts 0.5, g/L, concentration of herbicide 15 mg/L, and pH 5), 92% degradation of simeton was achieved in water samples.	[218]
	GO-Fe_3_O_4_	Reduced graphene oxide sheets were investigated using dopamine and decorated with magnetic Fe_3_O_4_ nanoparticles with an average size of 12 nm via a simple co-precipitation method, which produced artificial nano-enzymes. The results revealed that nano-enzymes were able to degrade 89% of simeton in water samples.	[207]
Terbutryn	TiO_2_	The photodegradation of terbutryn using titanium oxide nanotubes as a photocatalysis was studied. The results indicated that 70% of terbutryn was degraded within one hour.	[224]
	FeSO_4_-Fe (0)	The findings revealed that the FeSO_4_ and Fe (0) catalytic Fenton oxidation process was able to completely remove terbutryn and other pollutants from the wastewater.	[225]
	Zero-variant iron nanopowder	At various concentrations of terbutryn (1–1000 µg/L), complete removal of terbutryn was observed at pH 3 and 5 in wastewater using zero-variant iron nanopowder.	[226]
	N-TiO_2_	The results demonstrated that photocatalytic oxidation and ozonation using the highest tube diameter were efficient in removing almost 100% of terbutryn in aqueous samples.	[227]
Terbuthylazine	TiO_2_	Via a photocatalytic oxidation process, TiO_2_ nanoparticles effectively removed terbuthylazine from wastewater samples.	[228]
	FeSO_4_-Fe (0)	The findings revealed that the FeSO_4_ and Fe (0) catalytic Fenton oxidation process was able to completely remove terbuthylazine and other pollutants from the wastewater.	[225]
	TiO_2_	Titanium oxide nanoparticles were immobilized in chitosan glass fiber for the removal of terbuthylazine. Under UV irradiation, the satisfactory removal of terbuthylazine was observed in water samples.	[229]
	Zero-variant iron nanopowder	At various concentrations of terbutryn (1–1000 µg/L), complete removal of terbutryn was observed at pH 3 and 70% at pH 5 in wastewater using zero-variant iron nanopowder.	[226]
	TiO_2_-H_2_O_2_	The results demonstrated that under visible light and a hydrogen peroxide catalyst, titanium oxide nanoparticles efficiently performed and degraded 100% of terbuthylazine within 180 min.	[230]

The removal of three s-triazine herbicides (atrazine, propazine, and prometryn) from water by a thin film nanocomposite membrane containing oleic acid-modified silica nanoparticles was carried out through interfacial polymerization on a polysulfone asymmetric membrane. M-Phenylenediamine with trimesoyl chloride was polymerized to formulate a polyamide layer. Silica nanoparticles with a dimension of 15 nm were magnificently improved by oleic acid and merged into a trimesoyl chloride solution at concentrations ranging from 0 to 0.3 *w*/*v*%. The effects of specific physicochemical factors, such as weight and size, hydrophobicity, and the dipole moment of triazine molecules, on the membrane performance were investigated. The results showed that adding oleic acid-adapted silica nanoparticles to the polyamide membranes can increase the rejection and permeate flux. Prometryn and propazine had the highest and lowest rejection, respectively. The maximum rejection was achieved when the concentration of silica nanoparticles in the trimesoyl chloride solution was 0.17 *w*/*v*% [231].

## 5. In Silico Methods Used for Prediction of Toxicological Effects of S-Triazine Herbicides

Predicting the toxicological effects of s-triazine herbicides on different organisms is a critical aspect of assessing the safety of these chemicals. In silico methods, which involve computational techniques, have become increasingly valuable for toxicological predictions due to their cost-effectiveness and ability to complement experimental studies [232]. The most prevalent in silico techniques that are frequently used to forecast the toxicological effects of pesticides are quantitative structure–activity relationship (QSAR) analysis; toxicity prediction databases; toxicity pathway analysis; toxicokinetics and pharmacokinetics modeling; molecular docking and molecular dynamics simulations; in silico absorption, distribution, metabolism, excretion, and toxicity (ADME-Tox) models; and omics-integrated approaches [232,233,234]. QSAR models relate s-triazine herbicides’ chemical composition to their toxicological results in numerous organisms. To forecast toxicity, these models make use of characteristics derived from molecular structure, such as molecular weight, hydrophobicity, and electrical properties [235,236]. Recently, several classes of eight symmetrical triazine derivatives were compared in terms of their antifungal activity toward *Aspergillus flavus*, an opportunistic fungal pathogenic microorganism that frequently causes crops to become contaminated with aflatoxin, and in the context of their potential application as herbicides in maize, common wheat, barley, and rice crops. In this investigation, various techniques were applied, such as a chemometric pattern recognition approach (hierarchical cluster analysis), the experimental microbiological examination of antifungal activity (agar well-diffusion method), and molecular docking of the triazines in the appropriate enzymes. The primary results of the experiment indicate that the studied triazine derivatives have significant antifungal activity against *A. flavus*, especially those with acyclic substituents, and five of the eight studied triazines could be used as systematic herbicides, while the other three triazines could be used as contact herbicides. The compounds with acyclic substituents may be more appropriate for utilization in crop protection applications than triazines with cyclic substituents [237]. The (QSAR) technique was used to examine the attributes of four series of synthesized s-triazine derivatives that are important to pharmacokinetics and pharmacodynamics. Additionally, chromatographic retention and specific physicochemical characteristics linked to pharmacokinetics, such as (ADME-Tox), were connected. Furthermore, a principal component analysis (PCA) based on estimated ADME properties was carried out to confirm any similarities or differences across the series of tested substances. Standard statistical measurements and cross-validation parameters showed the statistical validity of the stated mathematical dependencies between retention parameters and ADME properties. The results of this study indicated that the s-triazine derivatives’ chromatographic retention values could be used in describing and assessing pharmacokinetic features [238]. To summarize, recent improvements in in silico techniques for predicting the toxicological effects of s-triazine herbicides have concentrated on enhancing the precision and thoroughness of toxicity evaluations [239,240]. As computational techniques continue to evolve, they will play an increasingly vital role in toxicological assessments and risk mitigation strategies [241,242].

## 6. Conclusions and Future Perspectives

S-triazine ring compounds are widely applied in various sectors as pesticides, resin intermediates, dyes, and explosives. Due to their unwise use, their residues are enormously increasing and causing serious hazards to non-target organisms, especially humans, mammals, pollinators, and predators across the globe. Various methods are applied to remove toxic residues and their intermediates from the environment. However, biodegradation methods play a pivotal role and are considered economic, eco-friendly, and more efficient. In the biodegradation method, various elements, including catabolic genes and enzymes, play a crucial role in catalyzing metabolic pathways and converting them into less toxic substances. More interestingly, in recent times, phytoremediation has been the most efficient technique to clean up various kinds of pollutants, including s-triazine compounds, from soil and water bodies to preserve natural habitats and ecosystems. Plants uptake toxic compounds through their roots, shoots, and leaves and transform them into less toxic or non-toxic compounds, which are further utilized by the plant system and participate in plant growth. In the past two decades, nanotechnology has contributed more efficiently to agriculture and other industrial sectors. Nanomaterials are considered superior agents for the complete removal of persistent pollutants due to their high range of versatility, adaptability, permeability, reusability, and high adsorption surface area. Additionally, the residues of s-triazine compounds induce public health and environmental issues; thus, nanomaterials are necessary for the efficient removal of pollutants in water bodies through adsorption, photocatalysis, sonocatalysis, sorption, ozonation, and biodegradation mechanisms. Instead of these practical approaches, more attention must be paid to the application of omics approaches, bioinformatic tools, database approaches, immobilization techniques, genetic tools such as genetic libraries and genetic fingerprinting, radio respirometry, and micro autoradiography for the complete removal of s-triazine compounds from the polluted environment. Furthermore, artificial intelligence and systematic biological methods are urgently required to inspect covered or invisible functional indigenous microbial communities.

## Figures and Tables

**Figure 1 microorganisms-11-02558-f001:**
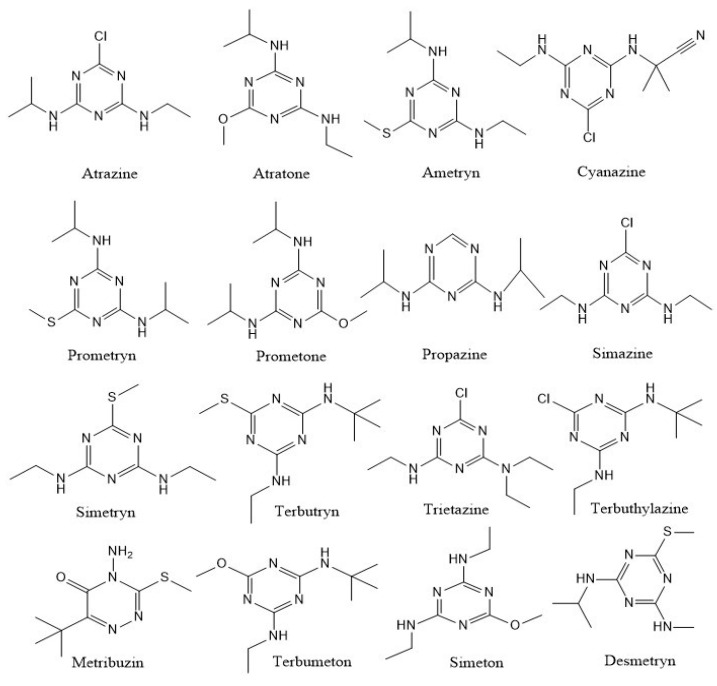
Representative compounds of S-Triazine herbicides.

**Figure 2 microorganisms-11-02558-f002:**
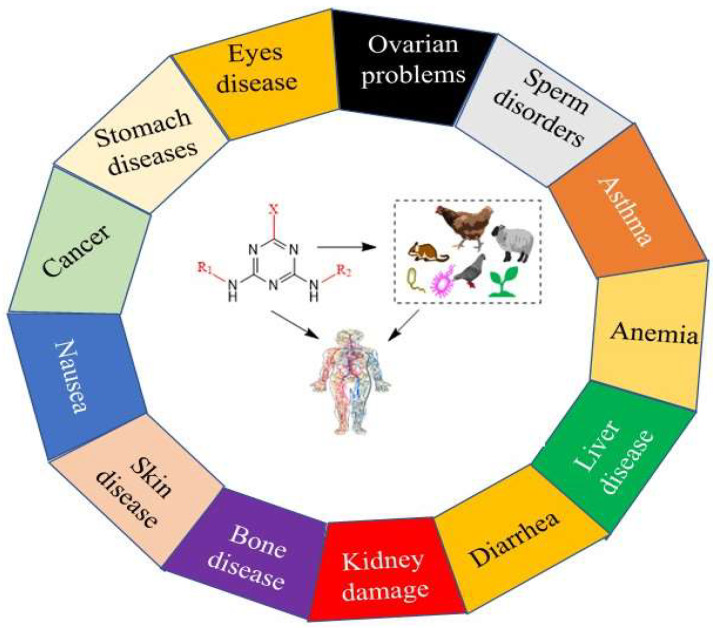
Toxicity of pesticides to non-target organisms.

**Figure 3 microorganisms-11-02558-f003:**
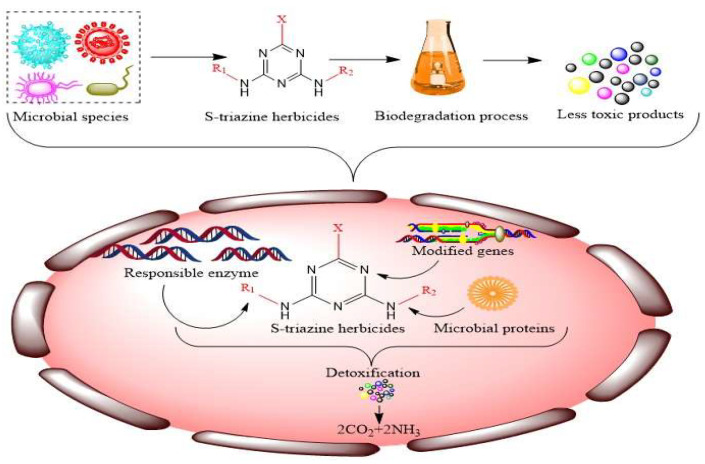
Microbial-mediated bio-purification.

**Figure 4 microorganisms-11-02558-f004:**
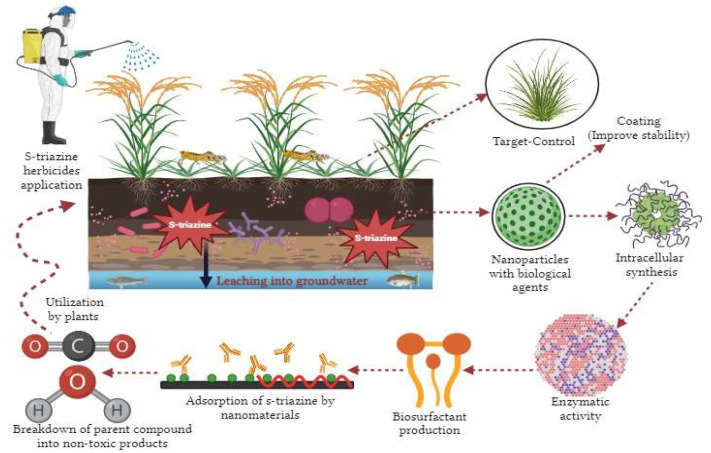
Nano-bioremediation of s-triazine herbicides.

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
