# Peer review of "Investigation of the Persistence, Toxicological Effects, and Ecological Issues of S-Triazine Herbicides and Their Biodegradation Using Emerging Technologies: A Review"

_microorganisms, 2023, doi:10.3390/microorganisms11102558_

Round 1
Reviewer 1 Report
I have reviewed the manuscript in which the authors review the degradation of s-triazine herbicides and their intermediates by indigenous microbial species, genes, enzymes, plants and nanoparticles have been systematically investigated. I propose to the authors to be more specific, explanatory and simplified in order to be easily understandable from the readers. The authors need to clarify a few points throughout the manuscript. Here are some suggestions to back up my idea and help the author improve the quality of the paper:
The title should indicate that it is a review.
Of particular relevance is a summary table on Table 1. Functional microbial species involved for the effective biodegradation of s-triazine herbicides. Or Table 2. Phytoremediation of s-triazine herbicides.
Some sentences are very long like the one between line 210 and 252. Please fragment.
In lines 276 FeSO4 and MnSO4 salts appear. Please put it properly.
Section 2.1 Degradation of s-triazines using microbial enzymes and genes is excessively long. I suggest to reduce between 10-20%.
It is necessary to follow the normal instructions of the journal in the References section
Author Response
Reviewer 1:
Dear Sir/Madam,
We appreciate the reviewer for putting a lot of time into reading, editing, and commenting on our manuscript. We are very much thankful for the critical comments and many constructive suggestions, which helped to improve the manuscript's quality. We have revised our manuscript based on the reviewer’s comments. Specific responses to the reviewers’ comments (in italics) are as follows:
Comment 1: I have reviewed the manuscript in which the authors review the degradation of s-triazine herbicides and their intermediates by indigenous microbial species, genes, enzymes, plants and nanoparticles have been systematically investigated. I propose to the authors to be more specific, explanatory and simplified in order to be easily understandable from the readers. The authors need to clarify a few points throughout the manuscript. Here are some suggestions to back up my idea and help the author improve the quality of the paper:
Response: Thank you for your comments and suggestions. As the reviewer suggested this manuscript was thoroughly revised. Corresponding changes were made in the revised manuscript.
Comment 2: The title should indicate that it is a review.
Response: As per reviewer suggestion, we included a review in the title. Corresponding changes we made in the line no. 3.
Comment 3: Of particular relevance is a summary table on Table 1. Functional microbial species involved for the effective biodegradation of s-triazine herbicides. Or Table 2. Phytoremediation of s-triazine herbicides.
Response: Thank you for your comments. As reviewer suggested we included summary for both tables. Corresponding changes were made in the revised manuscript at line no.201-206 and 567-571
Comment 4: Some sentences are very long like the one between line 210 and 252. Please fragment.
Response: As per reviewer suggestion, we revised the text. Corresponding changes were made in the revised manuscript.
Comment 5: In lines 276 FeSO4 and MnSO4 salts appear. Please put it properly.
Response: We modified.
Comment 6: Section 2.1 Degradation of s-triazines using microbial enzymes and genes is excessively long. I suggest to reduce between 10-20%.
Response: Thank you for your comments and suggestions. As per reviewer suggestions, we revised the section. Corresponding changes were made in the revised manuscript.
Comment 7: It is necessary to follow the normal instructions of the journal in the References section
Response: Thank you for your comments and suggestion. As per reviewer suggestion and Journal instructions. We modified the References section. Corresponding changes were made in the revised manuscript at line no.741-1364.
Reviewer 2 Report
The scientific names of organisms are not in italics. Grammar mistakes and typo errors are present. Please thoroughly review the whole paper.
Authors can discuss some in silico methods that can be used to predict the toxicological effects of s-triazine herbicides on different organisms. A comparison between the recent approaches can be included.
Fig 6. Can be improved.
Author Response
Dear Sir/Madam,
We appreciate the reviewer for putting a lot of time into reading, editing, and commenting on our manuscript. We are very much thankful for the critical comments and many constructive suggestions, which helped to improve the manuscript's quality. We have revised our manuscript based on the reviewer’s comments. Specific responses to the reviewers’ comments (in italics) are as follows:
Reviewer 2:
Comment 1: The scientific names of organisms are not in italics. Grammar mistakes and typo errors are present. Please thoroughly review the whole paper.
Response: Thank you for your comments and suggestions. As per reviewer suggestion, we changed scientific names of organisms in Italics. As the reviewer suggested this manuscript was thoroughly revised for grammar mistakes and typo errors. Corresponding changes were made in the revised manuscript.
Comment 2: Authors can discuss some in silico methods that can be used to predict the toxicological effects of s-triazine herbicides on different organisms. A comparison between the recent approaches can be included.
Response: Thank you for your comments and suggestions. As per reviewer suggestions, we include one section for in silico methods. Corresponding changes were made in the revised manuscript at line no.664-705.
Comment 3: Fig 6. Can be improved
Response: Thank you for your comments and suggestions. We included improved Figure 4 in the revised manuscript.